# The Potentiality of Rice Husk-Derived Activated Carbon: From Synthesis to Application

Md Masruck Alam [1], Md Ashraf Hossain [2], Md Delowar Hossain [1], M.A.H. Johir [3], Jewel Hossen [4], Md Saifur Rahman [1], John L. Zhou [3], A.T.M. Kamrul Hasan [1], Aneek Krishna Karmakar [1,*] and Mohammad Boshir Ahmed [1,3,5,*]

1  Department of Applied Chemistry and Chemical Engineering, University of Rajshahi, Rajshahi 6205, Bangladesh; masruckalam@gmail.com (M.M.A.); hdelowar5056@yahoo.com (M.D.H.); saifur_13bd@yahoo.com (M.S.R.); khasan@ru.ac.bd (A.T.M.K.H.)
2  Department of Materials Science and Engineering, Korea University, Seoul 02841, Korea; ashraf3521@gmail.com
3  Center for Green Technologies, School of Civil and Environmental Engineering, University of Technology Sydney, 15, Broadway, Sydney, NSW 2007, Australia; mohammed.johir@uts.edu.au (M.A.H.J.); junliang.zhou@uts.edu.au (J.L.Z.)
4  Department of Chemistry, Rajshahi University Engineering and Technology, Rajshahi 6204, Bangladesh; jewelhossenruet@gmail.com
5  School of Materials Science and Engineering, Gwangju Institute of Science and Technology, 261 Cheomdan-gwagiro (Oryong-dong), Buk-gu, Gwangju 500-712, Korea
*  Correspondence: aneek007@gmail.com (A.K.K.); Mohammad.Ahmed@uts.edu.au (M.B.A.)

**Abstract:** Activated carbon (AC) has been extensively utilized as an adsorbent over the past few decades. AC has widespread applications, including the removal of different contaminants from water and wastewater, and it is also being used in capacitors, battery electrodes, catalytic supports, and gas storage materials because of its specific characteristics e.g., high surface area with electrical properties. The production of AC from naturally occurring precursors (e.g., coal, biomass, coconut shell, sugarcane bagasse, and so on) is highly interesting in terms of the material applications in chemistry; however, recently much focus has been placed on the use of agricultural wastes (e.g., rice husk) to produce AC. Rice husk (RH) is an abundant as well as cheap material which can be converted into AC for various applications. Various pollutants such as textile dyes, organic contaminants, inorganic anions, pesticides, and heavy metals can be effectively removed by RH-derived AC. In addition, RH-derived AC has been applied in supercapacitors, electrodes for Li-ion batteries, catalytic support, and energy storage, among other uses. Cost-effective synthesis of AC can be an alternative for AC production. Therefore, this review mainly covers different synthetic routes and applications of AC produced from RH precursors. Different environmental, catalytic, and energy applications have been pinpointed. Furthermore, AC regeneration, desorption, and relevant environmental concerns have also been covered. Future scopes for further research and development activities are also discussed. Overall, it was found that RH-derived AC has great potential for different applications which can be further explored at real scales, i.e., for industrial applications in the future.

**Keywords:** rice husk; activations; adsorptions; dye; heavy metals; applications

## 1. Introduction

Activated carbon (AC) is a highly porous material, and it has versatile applications in environmental contaminant removal, electrode material preparation, development of supercapacitors, catalytic support for various applications, and energy storage system development [1–5]. AC has a high surface area,

a highly porous structure, and high thermal stability, as well as high acid and basic stability with different surface functional groups [6]. These properties develop when AC is produced through different physical or chemical activation processes. AC has widespread applications in the removal of inorganic material [7–9], organic pollutants [10,11], and gaseous environments [12], with applications with respect to energy. However, the versatile use of AC is sometimes hindered due to its high production and processing costs. Therefore, many attempts have been put forward for AC production from different low-cost precursors including industrial and agricultural waste [13–16]. Agricultural solid wastes are very common in every country, and are very cheap resources which can be converted into AC with excellent properties. Such kinds of examples are almond shells, hazelnut shells, poplars, walnut sawdust [17], orange peel [18], sawdust [19], rice husks (RHs) [20], sugarcane bagasse [21], coconut Burch waste [22], and tea leaves [23]. There are many precursors used to prepare AC. These include bituminous coal [24,25], wood [26], coconut shells [27,28], peat [29,30], petroleum pitch [31], and polymers [32]. In less developed countries, RHs are basically used as a heating source or to provide nutrients in soil, or are even dumped into the environment [33]. However, RHs can be converted into suitable materials before dumping them into the environment. Annually, 571 million tons of rice are produced, and 140 million tons of RH waste are generated [34,35].

The major components of RH are lignocellulose materials and mineral components. Besides, silica also becomes part of the RH during growth [36]. As a renewable and sustainable carbon resource, RH has been investigated in synthesizing AC materials for various applications [37,38]. Proper management of such cheap and abundant raw materials to produce valuable materials is of great importance. While different kinds of precursors have been utilized for AC production, low-cost production of AC is still a challenging problem [39,40]. Therefore, the utilization of RH for the production of AC can provide a better option with low-cost synthetic routes. In any case, to the best of our knowledge there is no literature review on RH-derived AC production and its potential for different applications and regeneration.

Henceforth, this review will investigate the properties of RH, activation processes for the synthesis of AC, and the potentiality of RH-derived AC for various applications. In addition, this review will provide a deep understanding of the regeneration, desorption of contaminants, and relevant environmental concerns of AC. Finally, this research will address the key relevant challenges for the future applications of RH-derived AC.

## 2. Properties of RH Precursor

RH is a form of lignocellulose biomass, and it is renewable waste, containing 28–30% inorganic and 70–72% organic compounds [41]. According to a previous study [41], the composition of the organic compounds includes C, H, O, N, and S (Table 1). The inorganic components are mainly constituted of silica [41]. RH has unique physicochemical and biochemical properties, which makes it a proper raw material for AC preparation [42]. During char formation, lignin works as the main constituent [43]. Cellulose and hemicellulose are associated with low carbon yields, and these volatile fractions removed during pyrolysis lead to the formation of microspore AC [44,45]. Generally, the properties of RH depend on several factors including geological location, rice variety, climate variation, cultivations methods, and fertilizers used in paddy growth [46–48]. For instance, the physical and thermochemical properties of RHs from Japan, Portugal, and Uganda were different [49–51]. Six different RH varieties from Bangladesh were thermogravimetrically analyzed by Ahiduzzaman et al. [52]. Table 1 highlights some additional studies on the geographical location of growth of RH according to physicochemical and biochemical properties, and an analysis of RH ash chemical compositions is provided in Table 2. However, RH has high ash content, which differs depending on geographical location.

**Table 1.** Physicochemical and biochemical properties of rice husk (RH) based on geographical location of growth (dry basis).

| Geographical Location | Biochemical Analysis (%) | | | Proximate Analysis (%) | | | Ultimate Analysis (%) | | | | | Ref. |
|---|---|---|---|---|---|---|---|---|---|---|---|---|
| | Lignin | Hemicellulose | Cellulose | Ash | Volatile Matter | Fixed Carbon | Carbon (C) | Hydrogen (H) | Oxygen (O) | Nitrogen (N) | Sulphur (S) | |
| China | - | - | - | 16.64 | 67.63 | 16.89 | 37.65 | 5.13 | 36.20 | 1.63 | 0.18 | [53] |
| Malaysia | 26.10 | 21.25 | 42.45 | 11.98 | 74.54 | 12.11 | - | - | - | - | - | [54] |
| India | - | - | - | 15.80 | 63.90 | 14.90 | 41.02 | 5.10 | - | - | - | [55] |
| Uganda | 10.58−13.47 | 11.39−19.97 | 31.3−36.54 | 15.87−25.56 | 58.78−63.37 | 14.77−17.75 | 29.98−34.48 | 4.46−5.59 | 40.48−43.36 | 0.36−0.63 | 0.005−0.041 | [49] |
| Bangladesh | - | - | - | 11.38 | 71.56 | 17.06 | 38.48 | 6.60 | 44.05 | - | - | [56] |
| Thailand | - | - | - | 11.97 | 72.80 | 9.30 | 48.90 | 6.20 | 44.10 | 0.80 | 0.30 | [57] |
| Portugal | - | - | - | 11.70 | 59.90−60.90 | 14.70−15.90 | 38.80−40.00 | 4.60−5.00 | 29.60 | 0.80−1.30 | - | [51] |
| Pakistan | 40.16 | 11.14 | 38.35 | 15.22 | 59.04 | 25.74 | 44.13 | 5.01 | 50.40 | 0.39 | 0.07 | [58] |
| South Korea | - | - | - | 12.98 | 73.73 | 13.28 | 55.13 | 6.43 | 38.43 | 0.01 | 0.00 | [59] |
| Japan | - | - | - | 12.70−22.00 | - | - | 35.2−41.2 | 4.6−5.4 | 48.8 | 0.4 | 0.6−0.7 | [50] |
| Egypt | 20.00 | 21.00 | 35.00 | 19.00 | - | - | - | - | - | - | - | [60] |

**Table 2.** Chemical composition of rice husk (RH) ash (total 100% in each column).

| Chemical Constituents | Weight (%) | | | | | | | | |
|---|---|---|---|---|---|---|---|---|---|
| | References | | | | | | | | |
| | [61,62] | [52,63] | [64] | [62] | [65] | [50,63] | [48] | [48] | [66] |
| Iron oxide ($Fe_2O_3$) | <0.5 | 0.12 | 0.2 | - | - | - | 0.09−0.27 | 0.09 | 0.26 |
| Aluminum oxide ($Al_2O_3$) | - | 0.11 | 0.41 | - | - | - | 0.09−0.25 | 0.05 | 0.39 |
| Calcium oxide (**CaO**) | 0.25 | 1.06 | 0.41 | 1.6 | 0.28 | 1.2 | 0.33−2.00 | 0.48 | 0.54 |
| Magnesium oxide (**MgO**) | 0.23 | 0.33 | 0.45 | - | 0.14 | - | 0.30−0.45 | 0.44 | 0.90 |
| Manganese oxide (**MnO**) | - | 0.08 | - | - | 0.03 | - | - | - | 0.16 |
| Silica ($SiO_2$) | 94.5 | 95.79 | 96.34 | 91.1 | 95.1 | 91.5 | 90−97 | 96.0 | 94.95 |
| Sodium oxide ($Na_2O$) | 0.78 | 0.30 | 0.08 | - | - | - | 0.03−0.23 | 0.08 | 0.25 |
| Potassium oxide ($K_2O$) | 1.10 | 2.17 | 2.31 | 5.3 | 0.13 | 4.3 | 1.80−2.80 | 2.10 | 0.94 |
| Phosphorous pentaoxide ($P_2O_5$) | 0.53 | - | - | 0.6 | - | 1.1 | 0.03−1.20 | 0.59 | 0.74 |
| **Others** | - | 0.04 | - | 1.4 | - | 1.9 | - | - | - |

## 3. Pretreatment, Potentiality, and Drawbacks of Precursors and Synthesis of AC

### 3.1. Pretreatment of RH Feedstock

RH is usually composed of lignocellulosic materials together with a high amount of silica and other metallic content. Table 1 shows the physicochemical properties of RH precursors of different origins, and Table 2 highlights the properties of RH ash. From Table 1, it can be clearly seen that the properties of RH vary by the geographical location [46]. It is reported that a high amount of ash content should not be present in precursors as it can hinder the pore development during further treatment i.e., AC production [67]. During activation, the activating agent can also react with the silica which is present in the RH. As a result, the surface area of AC produced from lignocellulosic materials with low ash content becomes higher than in the raw RHs under similar preparation conditions [68]. Leaching of RHs by acids or bases is, therefore, essential to escape the detrimental influences of the unconventionally high ash content in the RH [42,67,69,70]. It has been reported that the surface area and pore widening increased on leaching of RH [1]. Alkaline pretreatment of RHs by the utilization of 2–4% *w/v sodium hydroxide* (NaOH) could reduce the ash content up to 74–93% [71]. According to Equation (1), leaching of RHs by employing a base is highly possible [1].

$$2NaOH(s) + SiO_2(s) \rightarrow Na_2SiO_3(s) + H_2O \tag{1}$$

It is easy to remove water-soluble sodium silicate ($Na_2SiO_3$) by washing with water [1]. Figure 1 shows an overview of preleaching and AC production from RH. In addition, most of the remaining metallic impurities in the RHs can be removed by acid-leaching of RHs, which can further hinder pore development [71]. Liou et al. [72] reported that when RHs were refluxed with 3 N hydrochloric acid (HCl) at 100 °C for 1 h, about 84% of the metallic impurities were extracted. Deiana et al. [47] found a process that is best suited for the leaching of impurities. The process followed the sequence of carbonization, activation, and leaching, respectively, to get the desired properties.

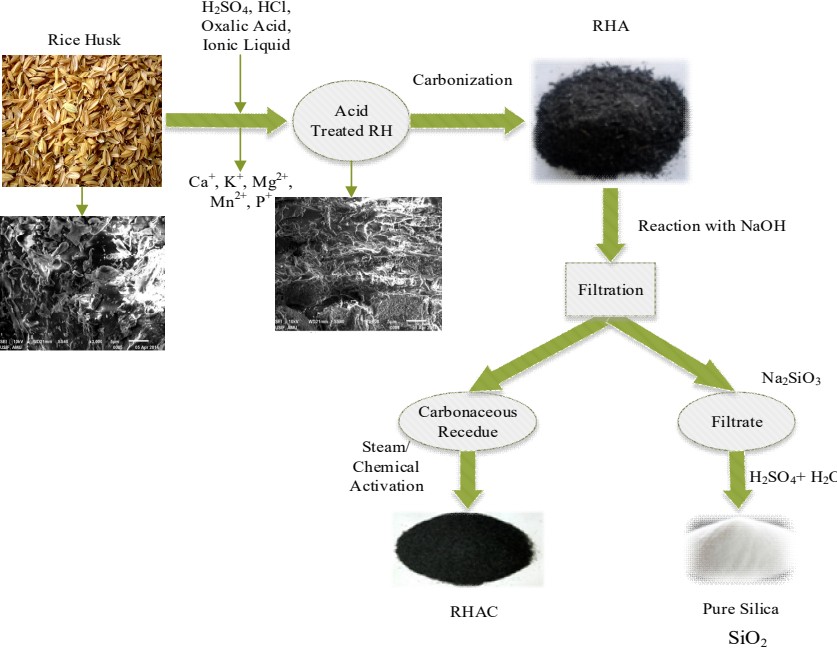

**Figure 1.** Overview on the preclearing and activation processes of Rice husk (RH) for activated carbon (AC) production. RH ash (RHA): RH ash. In figure, $Ca^{2+}$, $K^+$, $Mg^{2+}$, $Mn^{2+}$, P and $H_2SO_4$ indicates calcium ion, potassium ion, magnesium ion, manganese ion, phosphorous, and sulfuric acid, respectively. Reproduced with permission from [1,47,72]. Elsevier and Copyright Clearance Center, 2009; Copyright © 2008, & Copyright © 1997, American Chemical Society, respectively.

On the other hand, silica needs to be removed for specific applications in order to increase the quality of RH for the production of AC. In that case, base leaching can be effective, as it removes the main constituent lignin from the RH as well as silica, leading to a low yield of carbon (reaction 1). For example, Yeganeh et al. [67] reported a lower yield of carbon with base leaching compared to non-leaching of RH. However, the leaching system ought to be carefully carried out in a way that does not adversely compromise the overall yield and properties of the RH-derived AC.

### 3.2. Potentiality and Drawbacks of Modified RH

RH is considered to be an effective adsorbent for the removal of a range of pollutants. RH has many advantages over other precursors. For example, it has high amounts of cellulose and carbon compared to many woody biomasses. There is no need for extra steps for biomass processing (e.g., cutting, sizing, and so on), and it provides higher AC yield even with high temperatures. It has high surface area and high adsorption capacities which are comparable to those of commercial AC (Tables 1 and 3). The use of RH or another type of biomass has two major drawbacks, namely, physical instability and low sorption capacity [73]. Hence, it is necessary for RH to go through proper modification treatments for removing metallic impurities, lignin, and other easily accessible functional groups from RHs as well as silica in order to overcome the associated problems [74].

RH biomass can be modified physically and chemically. Sometimes, only heating is very effective. For example, when RH biomass was heated at 100 °C and applied for biosorption, increased biosorption capacity was observed due to the denaturation of lignin [75]. The adsorption capacity can be improved by the modification of the biomass with HCl, $H_2SO_4$, and nitric acid ($HNO_3$). Modification of RH biomass with salts (e.g., sodium chloride-NaCl, calcium chloride-$CaCl_2$, and manganese sulphate-$MgSO_4$) can also increase the adsorption capacity. For example, an increased biosorption capacity of dye has been observed [76]. HCl treatment of RH biomass can significantly increase the biosorption capacity [77]. Janoš et al. [78] treated the RH with HCl, sodium carbonate ($Na_2CO_3$), and sodium by phosphate ($Na_2HPO_4$). The results showed that the biosorption capacity of methylene blue was increased after treatment with $Na_2CO_3$. On the other hand, Hsu et al. [79] reported 14 times higher adsorption capacity of methacrylic acid on modified RH. Ong et al. [80] mentioned that ethylene diamine tetra acetic acid (EDTA)-modified RH was capable of acting as a single sorbent to remove both methylene blue and Reactive Orange16 dyes. Hence, modification of RH can increase the adsorption or biosorption capacity of different pollutants.

Therefore, instead of pristine RH, modified RH can have better potential to enhance the sorption or biosorption capacity of different pollutants. However, further chemical or physical treatments are necessary in order to boost its adsorption capacity to reach a level similar to that of commercial activated carbon (CAC). Hence, further treatments are necessary.

### 3.3. Activation of RH for AC Synthesis

Various chemical, physical, physicochemical, and pyrolysis processes are utilized for the synthesis of AC from RH. The physical process includes physical treatment of RH biomass at high temperatures in the presence of steam or carbon dioxide. On the other hand, chemical treatment includes the utilization of activating agents through impregnation followed by activation at a higher temperature under inert environments [1,50,81–84]. However, the preparation of AC from RH by the utilization of combination of these methods known as "physicochemical activation" has also been successfully employed [85]. Figure 2 illustrates the preparation of AC through different methods.

**Table 3.** Physiochemical characteristics of the rice husk (RH)-derived activated carbon (AC).

| Production Temperature °C | Activating Agent | Brunauer-Emmett-Teller (BET) Specific Surface Area (m²/g) | Total Pore Volume (cm³/g) | Ash Content (%) | Yield (%) | Ref. |
|---|---|---|---|---|---|---|
| 650 | KOH | 280 | 0.206 | 42.6 | 33.4 | [61] |
| 600 | CaCl$_2$ | 171 ± 1 | - | 40.5 | - | [87] |
| 200 | HNO$_3$ + K$_2$CO$_3$ | 542 ± 2.3 | - | 17 | - | [88] |
| 900 | H$_3$PO$_4$ | 438.9 | 0.3871 | - | 37.69 | [89] |
| 900 | Na$_2$CO$_3$ + K$_2$CO$_3$ | 1581 | 1.44 | - | - | [63] |
| 700 | ZnCl$_2$ | 750 | 0.38 | 2.0 | 17.7 | [86] |
| 650 | ZnCl$_2$ | 180.50 | 2.70 | 25.7 | - | [90] |
| 900 | H$_3$PO$_4$ | 420 | - | - | 37.69 | [89] |
| 100 | H$_2$SO$_4$ | 681 | 0.526 | -3.0 | 36 | [91] |
| 300 | ZnCl$_2$ | 578 | 0.463 | 3.3 | 32 | [91] |
| 600-800 | NaOH | 1400 | - | - | - | [90] |
| 600 | H$_2$SO$_4$ | 17.2 | 0.48 | 7.80 | - | [92] |
| 850 | Water vapor | 1180 | 1.09 | 1.0 | 6.9 | [47] |
| 850 | CO$_2$ | 334 | 0.207 | 61.5 | 33.2 | [50] |
| 850 | CO$_2$ | 460 | 0.261 | 47.7 | 26.1 | [50] |
| 850 | CO$_2$ | 388 | 0.231 | 55.1 | 30.60 | [50] |
| 850 | CO$_2$ | 325 | 0.200 | 60.7 | 34.2 | [50] |
| 850 | CO$_2$ | 473 | 0.267 | 52.2 | 26.8 | [50] |
| 800 | KOH | 3014 | 1.73 | - | - | [93] |
| 750 | NaOH | 2952 | 1.88 | - | - | [93] |
| 750 | NaCO$_3$ | 600 | 0.286 | 9.7 | - | [93] |
| 750 | K$_2$CO$_3$ | 1100 | 0.536 | 6.2 | - | [93] |
| 875 | CO$_2$ | 466.9 | 0.35 | 58.8 | 26.5 | [94] |
| 600 | Steam | 272.5 | - | 45.97 | - | [82] |
| 500 | H$_3$PO$_4$ | 352 | 0.4158 | 3.3 | 33 | [95] |

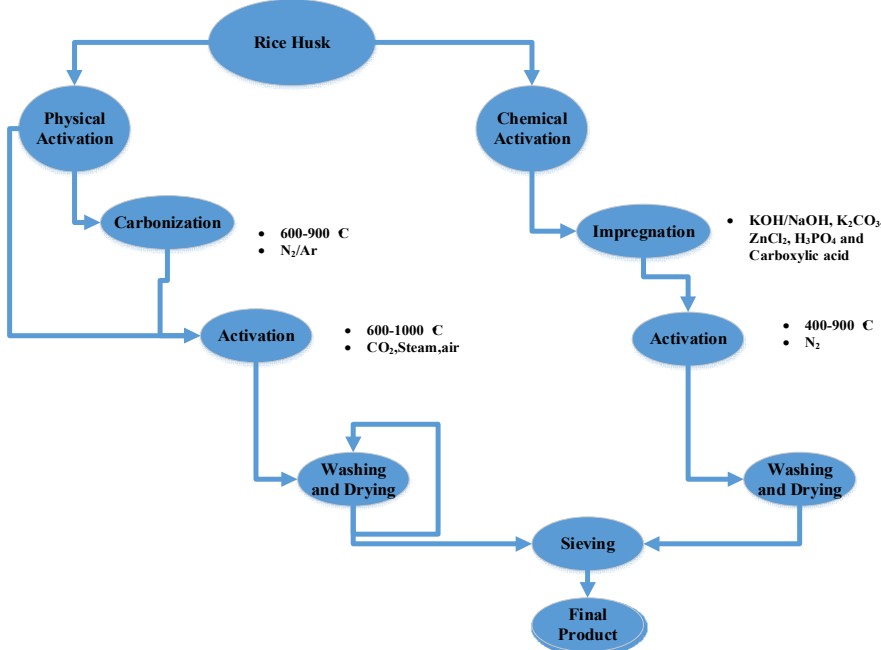

**Figure 2.** Outline of AC preparation from RH. In figure, KOH, ZnCl$_2$, H$_3$PO$_4$, and N indicate potassium hydroxide, zinc chloride, phosphoric acid and nitrogen, respectively. Reproduced with permission from [38]; Copyright © 2012, American Chemical Society.

### 3.3.1. Physical Method and Thermal Activation of RH for AC Production

The adsorption capacity of the parent material can be improved by physical modification using carbon dioxide and water vapor. On the other hand, thermal processes use high temperatures for the production of highly porous carbons. RH/biomass treatment is carried out at high temperatures (up to ~1200 °C) with steam, carbon dioxide, or a mixture of these through physico-chemical processes. Both steam and carbon dioxide are mild oxidants, and they can depolymerize and fragment the biomass/RH into lower carbons [47,82,86]. However, the resulting carbon has a higher adsorption capacity than pristine RH. Subsequently, during the carbonization process, new porosity is created along with a high surface area in the AC materials (Figure 3). However, the weight of the host carbon decreases due to loss of carbon during activation. Physical/physico-thermal activation to prepare AC from RH involves low specific surface areas because of high silica content [47]. Data on the activation of RH using steam and carbon dioxide are shown in Tables 3 and 4.

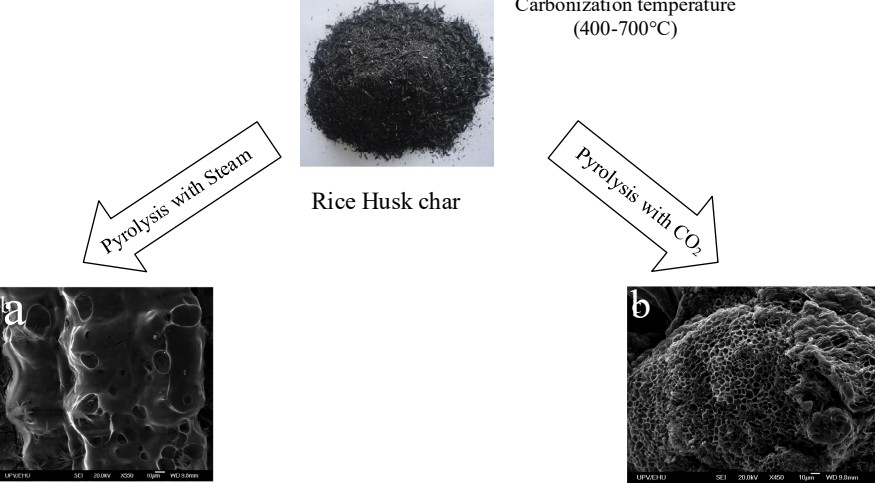

**Figure 3.** Physical activation of RH. (**a**) SEM image of RH-derived AC with steam; (**b**) SEM image of RH-derived AC with $CO_2$. Reproduced with permission from [81]; Copyright © 2015, American Chemical Society.

Table 4. Detailed physical activation processes for AC production from RH.

| Pre-Treatment of RH | Pyrolysis Conditions | Chemical Agents | Activation Conditions | Surface Area $(m^2/g)$ | Ref. |
|---|---|---|---|---|---|
| Base-leached | | Steam | Temperature 946 °C, heating rate 8 °C $min^{-1}$, activation time 31 min | 1004.30 | [96] |
| Not leached | Temperature 400 ± 10 °C for 1 h under $N_2$ flow | Steam (50% $H_2O$/50% $N_2$) | Temperature 800 ± 10 °C, heating rate 3.5 °C $min^{-1}$, steam flow rate 0.4 NL $min^{-1}$, activation time 10–240 min | 362.30 | [97] |
| No leaching | Temperature 400 °C for 1 h under $N_2$ atmosphere | Steam | Temperatures 600 °C, heating period 1 h, steam pressure 1.5 kg $cm^{-2}$ | 272.50 | [82] |
| Acid-leached | Temperature 500 °C, $N_2$ flow 1.4 K $min^{-1}$, heating time 2 h | Steam | Heating rate 5 K $min^{-1}$, temperature 850 °C in $N_2$ gas flowing at 300 mL min−1, switch to steam for 105 min | 1180.0 | [47] |
| Not-leached | Temperature 400 °C, heating period 1 h, $N_2$ flow 500 mL $min^{-1}$ | $CO_2$ | Temperature 800–900 °C for 1 h and at 850 °C for 0.25–3 h in a $CO_2$ flow of 500 mL $min^{-1}$ | 325–473 | [50] |
| Acid-and base-leached | RH char obtained by flash pyrolysis at 500 °C | Steam/$CO_2$ | Temperature 800 °C, heating rate 15 °C $min^{-1}$, activation time 15–60 min. | 1514.0 | [81] |
| Not leached | RH, $N_2$ flow rate 35 mL $min^{-1}$, heating rate 20 °C $min^{-1}$ | $CO_2$ | $CO_2$ flow rate of 200 mL $min^{-1}$, activation temperatures 650, 750 and 850 °C | 350.1 | [98] |
| Not leached | RH pyrolyzed at 700 °C for 30 min under $N_2$ atmosphere | $CO_2$ | Activation temperature 700 °C, activation time 30 min | 166.0 | [86] |

### 3.3.2. Chemical Activation of RH for AC Production

Chemical activation refers to the impregnation of raw materials with various chemical agents followed by the high-temperature heat treatment in the presence of the inner atmosphere or inert gases. This process is often employed where the precursor consists of cellulosic or organic and inorganic materials. $ZnCl_2$, $H_3PO_4$, sodium by carbonate ($NaHCO_3$), KOH, NaOH, and $H_2SO_4$ are commonly used reagents for the impregnation (Table 3). This process requires the reaction of the biomass with the activating agents, and the application of high temperatures ranging from 500 °C to 1000 °C (Figure 2).

Different activating agents have different roles as well as different mechanisms in the activation of biomass, which can result in the further development of well-defined structures within the material [42,83,86]. The main mechanism is that activating agents are capable of penetrating inside the biomass, and can then break down the cross-linkages of cellulosic materials. There are two types of processes. One method is the activation of biomass followed by carbonization directly [83,86]. The second is a two-stage process in where the precursor material is pre-carbonized and subsequently chemically activated followed by carbonization at high temperatures [84,99]. The second process has added advantages. For instance, a surface area of 280 $m^2$/g was obtained through NaOH activation and carbonization of RH, and a surface area of 660 $m^2$/g was obtained from NaOH activation of pre-carbonized RH followed by carbonization [83,100]. Proper selection of the activation process is also necessary. For example, concentrated $H_2SO_4$ is a strong oxidizing agent, and oxidation through dehydration occurred in the cellulose and hemicelluloses [101]. In that sense, $ZnCl_2$ is the most preferable activating agent for the chemical activation of lignocellulosic materials [102]. On the other hand, phosphoric acid is not linked with problems of aggressive corrosion, chemical recovery, and other environmental disadvantages that are associated with other activating agents such as $ZnCl_2$. Tables 3 and 5 show different experimental data for chemical activation. Hence, to get the best results it is necessary to maintain optimum conditions including pre-carbonization conditions, impregnation ratio, type of activating agent, heating rate, activation temperature, activation time, and atmosphere. Therefore, these factors need to be optimized to get the desired adsorption properties of produced AC.

### 3.3.3. Comparison between Physical and Chemical Activation

There are many properties that are comparable between physical and chemical activations of biomass under certain conditions. For example, the chemical activation process utilizes activating agents which add additional costs to the synthetic process, while the physical process does not consume costly chemicals [86,103]. Conversely, the surface area of physically activated AC is lower than that of chemically activated carbon (Table 5) [18,86,104]. For example, surface areas of 750 and 166 $m^2$/g were obtained by employing chemical and physical activation of RHs, respectively, under the same activation temperature [86]. The yield of AC through chemical activation is lower than that of the physical process, but fine porous structures can be obtained through chemical processes. Nevertheless, chemical activation is related to problems of corrosion and needs for washing of AC along with the problem in activating agent recovery [103,105]. Therefore, both processes have some advantages and disadvantages in comparison to each other. However, commercially chemical AC is widely used due to its high adsorption and specific capacity [81,106,107].

**Table 5.** Adsorption performance of physically and chemically activated RH-derived AC.

| Adsorbate | Physical Activation | | | Chemical Activation | | | Ref. |
|---|---|---|---|---|---|---|---|
| | Pore Volume (cm$^3$/g) | BET Surface Area (m$^2$/g) | Adsorption Capacity or Removal Percentage | Pore Volume (cm$^3$/g) | BET Surface Area (m$^2$/g) | Adsorption Capacity or Removal Percentage | |
| Methylene blue | 0.37 | 417 | 28.5 mg/g | - | 143 | 44.25 mg/g | [108] |
| Malachite Green | 0.068671 | 9.8 | 97.3% | 0.027 | 180.50 | 94.91% | [109,110] |
| Acid Yellow 36 | - | 272.5 | 86.9% | - | - | 100% | [82,111] |
| Phenol | - | - | 90% | 1.126 | 1836 | 75.0 | [112,113] |
| Humic acid | - | - | 98.24% | | 724 | 74% | [114,115] |
| Fe (III) | - | - | 99% | 0.3316 | 994.32 | 100% | [116] |
| Ni (II) | | | 18.4 ± 1.4% | 0.42 | 1563 | 92.6% | [117,118] |
| Cd (II) | - | - | 125.94 mg/g | | 14 | 28.27 mg/g | [119,120] |
| Pb (II) | 0.23 | 610.1 | 111.9 mg/g | 0.38 | 1038.6 | 236.2 mg/g | [121] |
| Hg (II) | - | - | 4.0 mg/g | 0.835 | 2786 | 342.0 mg/g | [122,123] |
| As (III) | - | - | 96–100% | 0.43–0.57 | 811–1624 | 1.22–1.32 mg/g | [124,125] |
| Zn (II) | - | - | - | 0.41 | 604.34 | 40.87% | [126] |
| RO-16 | | | | - | - | 13.32 mg/g | [127] |
| Cr (4) | - | 380 | 94% | - | 571.07 | 152.91 mg/g | [128,129] |
| Mn (II) | - | - | 98% | 0.3316 | 994.32 | 100% | [116,130] |

### 3.3.4. Hydrothermal Treatment of RHs

Recently, hydrothermal treatment has become more popular as it results in high yields of carbons, the production of quality products, the removal of minerals, and use of less energy compared to other thermal processes. In comparison, hydrothermal/thermo-chemical treatment can transform different biomasses at lower temperature (<200 °C), with high pressure and high moisture content in the feedstock. Using this process, biomass can easily be hydrolyzed and dehydrated [131]. Hydrothermal treatment can be used for adsorbent preparation, catalysis synthesis, materials for water purification, materials for energy storage, and $CO_2$ sequestration [132,133]. A number of studies have reported the hydrothermal conversion of RHs. For example, hydrothermal treatment has been applied for hydrochar synthesis [134], cellulose nanofiber synthesis [135], and materials for Li-ion batteries [132], among other applications. Direct synthesis of AC from RH precursors using hydrothermal treatment has not been explored yet. Therefore, this process needs further consideration in the case of AC production from RH.

### 3.3.5. Cost-Effective Synthesis of AC

Many attempts have been put forward to reduce the cost of AC production from different biomasses. Therefore, special emphasis has been placed mainly on low-cost precursors. In that case, agricultural wastes (e.g., RH) have been widely used for producing AC. Recently, a green preparation technology has been proposed based on the comprehensive utilization of RH which eliminates the potential waste generation and emissions to the environment [36]. After the activation and carbonization of biomass, much waste is produced, which can be very harmful to the environment [136,137]. The cost-effective synthesis technology almost did not produce pollutants, as shown in Figure 4. Also, the activating agent and water used were recycled. Hence, the entire technology is green and environmentally benign.

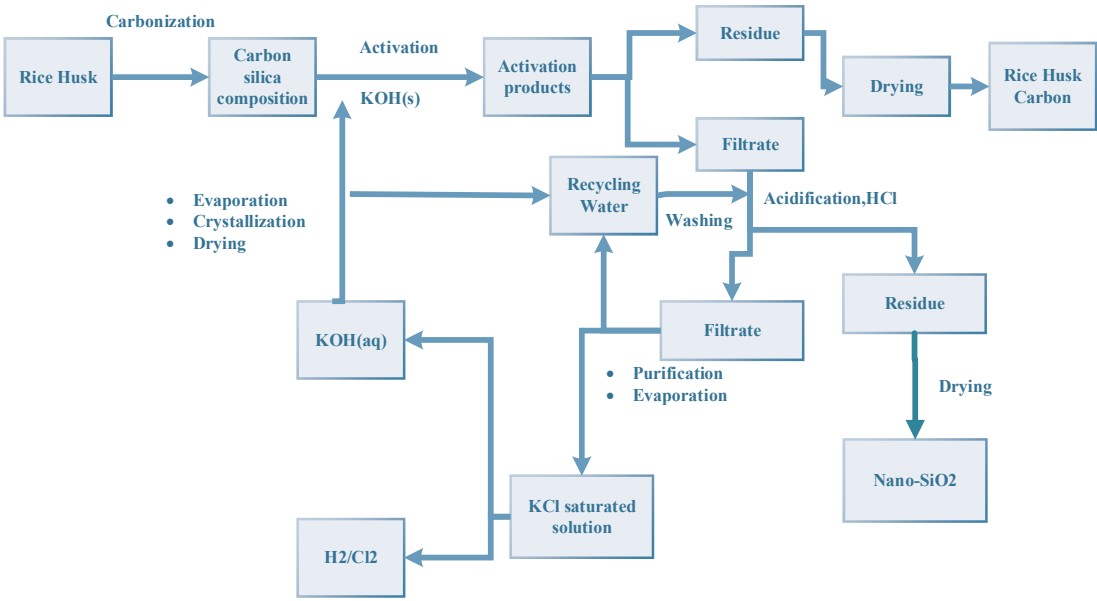

**Figure 4.** Schematic representation of cost-effective AC production technology. Reproduced with permission from [36]; Elsevier and Copyright Clearance Center, 2016.

On the other hand, very recently Ahmed et al. [138] proposed a facile method of producing AC with lower chemical consumption. They mentioned that about a 70% reduction in chemical requirements can be achieved by the utilization of a technique which includes a prepyrolysis of the biomass before chemical activation of biomass instead of the direct chemical impregnation of biomass. Consequently, this technology is very economical and environmentally friendly in comparison with other technologies which consume huge amounts of activating agents while also producing large amounts of liquid waste.

## 4. Environmental Applications, Mechanisms, Regeneration, Desorption, and Environmental Concerns of RH-Derived AC

Carbon-based materials, which are highly porous, can play a critical role in adsorption applications (contaminants and gas adsorption) as well as in catalytic support materials, capacitors, battery electrodes, supercapacitors, and gas storage. To improve performance, AC should have high surface area together with mesopore or macropore structures [89]. In this section, environmental applications through the adsorption processes of different compounds and contaminants as well as different types of gas capture by RH-derived AC will be discussed.

### 4.1. Adsorption of Organic Pollutants

### 4.1.1. Adsorption of Dyes, Phenolic Compounds, and Others

Every day many organic contaminants are disposed through different routes. They are eventually mixed with environmental matrices (e.g., soil, sediments, and water). A significant proportion remains unchanged in the environment. Therefore, they can pose a significant threat to our health and aquatic systems [139]. Among different routes, textile industries are dumping huge amounts of organic dyes and pigments. They consume large amounts of water and dying chemicals, and subsequently discharge huge amounts of effluents with synthetic dye and pigments into the environment, causing public health concerns. Numerous methods can be adopted for the treatment of industrial dyes from wastewaters. Among them, the adsorption process is the most effective and attractive method for the treatment of organic contaminants, as the adsorbents are inexpensive and they do not require extra pretreatment before their application. Satisfactory results were obtained with respect to the removal of many organic chemicals by the utilization of AC as an adsorbent [140–142]. Among different adsorbents, AC is heavily applied for the adsorption of different organic contaminants.

Among different kinds of AC, RH-derived AC has many advantages including high abundance, cost-effectiveness, and forms of mesoporous or microporous AC that can be prepared from RHs [143]. The adsorption capacities of various dyes by RH, modified RH, and RH ash (RHA) are shown in Table 6. The oxalic acid-modified RH was utilized for the adsorption of methylene blue and malachite green. It was found that the adsorption capacity of modified RH increased from 19.77 to 53.21 mg/g and 28.00 to 54 mg/g at 293 K, respectively, for methylene blue and malachite green [144]. In one study RH dried at 60 °C for 48 h showed maximum adsorption for safranine and methylene blue 760 mg/g and 280 mg/g, respectively [145]. It should be taken into consideration that commercial activated carbon (CAC) has a higher adsorption capacity for organic pollutants. However, the cost of CAC is several times higher than that of other low-cost precursor-derived ACs.

**Table 6.** Organic contaminant adsorption capacity by AC prepared from RH and pristine RH. CAC: commercial activated carbon.

| Adsorbent | Activation | $q_e$ (mg/g) | Adsorbate | Ref. |
|---|---|---|---|---|
| RH | Oxalic acid/20 °C | 28.00 | Malachite green | [144] |
| RH | Dried at 60 °C | 25.63 | Direct Red-31 | [146] |
| RH | - | 19.96 | Direct Orange-26 | [147] |
| RH | Dried at 60 °C | 760 | Safranine | [145] |
| RH | Steam/100 ± 5 °C | 86.9 | Acid Yellow 36 | [82] |
| RH | NaOH/400 °C | 511 | Basic Green 4 | [148] |
| RH | Dried at 60 °C | 280 | Methylene Blue | [145] |
| RH | Dried at 80 °C | 838 | Basic Red 2 | [149] |
| RH | Dried at 60 °C | 14.00 | Congo Red | [147] |
| RH | Dried at 80 °C | 178.10 | Safranine | [149] |
| RH | Dried at 80 °C | 312 | Methylene blue | [149] |
| Oxalic acid modified RH | Oxalic acid/20 °C | 53.21 | Methylene blue | [144] |
| EDTA-RH | NaOH/70 °C | 46.30 | Methylene blue | [80] |
| Oxalic acid-modified RH | Oxalic acid/20 °C | 54.02 | Malachite green | [144] |
| NaOH-modified RH | NaOH/70 °C | 17.98 | Malachite green | [150] |
| EDTA-RH | NaOH/70 °C | 7.68 | Reactive Orange 16 | [80] |
| HCl-RH | Steam/100 °C | 50.0 | Direct Blue 67 | [151] |
| Carboxy methyl cellulose-RH | Steam/100 °C | 50.0 | Direct Blue 67 | [151] |
| AC | Steam/900 °C | 19.2 | Methylene blue | [152] |
| AC | $CO_2$/900 °C | 20.2 | Methylene blue | [152] |
| AC | Steam + $CO_2$/ 900 °C | 20.8 | Methylene blue | [152] |
| AC | Steam/900 °C/ $O_3$ at room temperature | 24.8 | Methylene blue | [152] |
| AC | $CO_2$/900 °C/ $O_3$ at room temperature | 26.2 | Methylene blue | [152] |
| AC | Steam + $CO_2$/900 °C/$O_3$ at room temperature | 27.8 | Methylene blue | [152] |
| AC | Steam/700 °C | 19.89 | Methylene blue | [153] |
| AC | $H_2O_2$/400 °C | 26.6 | Malachite green | [154] |
| AC | $HNO_3$/400 °C | 18.1 | Malachite green | [154] |
| AC | $HNO_3$/400 °C | 14.1 | Methylene blue | [155,156] |
| AC | $H_2O_2$/400 °C | 18.7 | Methylene blue | [155,156] |
| AC | NaOH/70 °C | 9.8 | Malachite green | [157] |
| AC | $H_2O_2$/110 °C | 13.2 | Safranin-T | [140] |
| CAC | Thermal | 298.4 | Methylene blue | [153] |
| CAC | - | 490 | Mordant Blue-9 | [158] |
| CAC | - | 200 | Methylene blue | [159] |

Phenol is an important organic pollutant which gives water an unpleasant taste and odor when dumped into the environment. It is a semi-volatile organic compound commonly present in oil refinery wastes. Phenol is also produced during the transformation of coal into gaseous and liquid fuels. There are many other sources of phenol pollution. These mainly include paint manufacture, coal conversion, pesticides, petrochemicals, and industrial polymeric resins [160–162]. AC can remove phenol from water to a great extent. The adsorption capacities of organic compounds onto RH and RH ash (RHA) together with phenols are listed in Table 7. So far, most of the experiments on the phenol adsorption by different ACs have been conducted on a solution basis. However, the adsorption of phenol in the gas phase has also been reported. For example, porous carbons produced from the unaltered and pelletized RH showed a comparatively high adsorption capacity (191.9 mg/g) of phenol in the vapor phase. This was due to the presence of the main silica in the RH [112,163]. RH-derived AC has also been reported to remove phenol, with an adsorption capacity of 201 mg/g within a few minutes of adsorption [164]. Hence, it can be assumed that silica has a great influence on phenolic compound removal.

**Table 7.** Adsorption capacities of phenols and other organic compounds by pristine rice husk (RH), RH ash (RHA), and RH-derived AC.

| Adsorbent | Adsorbate | Adsorption Capacity (mg/g) | Ref. |
|-----------|-----------|----------------------------|------|
| RH | Phenol | 191.9 | [112] |
| RH | Phenol | 4.508 | [165] |
| RH | Phenol | 201 | [164] |
| RH | p-Chlorophenol | 14.36 | [165] |
| RH | p-Nitro phenol | 15.31 | [165] |
| AC | Paraquat | 317.7 | [79] |
| AC | Phenol | 7.91 | [165] |
| AC | p-Chlorophenol | 36.23 | [165] |
| AC | p-Nitrophenol | 39.21 | [165] |
| CAC | 2,4-Dichlorophenol | 48.18 | [166] |
| RHA | Phenol | 0.886 | [167] |
| RHA | Phenol | $143.99 \times 10^{-4}$ | [168] |
| RHA | phenol | 0.951 | [169] |
| Granular AC | Phenol | 1.00 | [169] |
| RHA | Phenol | 0.989 | [169] |
| RHA | Resorcinol | $888 \times 10^{-5}$ | [168] |
| RHA | 2-Chlorophenol | $209.55 \times 10^{-6}$ | [168] |
| RHA | Pyridine | 11.72 | [170] |
| RHA | $\alpha$-Picoline | 15.46 | [171] |
| RHA | Humic acid | 2.7 | [172] |
| AC | Phenol | 27.58 | [173] |
| AC | Humic acid | 8.2 | [172] |
| AC | Humic acid | 21-45 | [174] |
| CAC | 2,4-Dichlorophenol | 48.18 | [165] |
| CAC | Chloramphenicol | 343 | [175] |

On the other hand, H$_3$PO$_4$-impregnated RH and RH char were utilized for the removal of p-nitro phenol, where a maximum adsorption capacity of 39.21 mg/g was reported [165]. CAC was also utilized

for the removal of phenolic compounds. However, their adsorption capacity was not significant as that of RH-derived carbons. For instance, the adsorption capacity of 2,4-dichlorophenol by $H_3PO_4$-modified CAC was found to be 48.18 mg/g [166]. Therefore, RH-derived carbons have great potential in the removal of phenolic compounds as compared to CAC.

### 4.1.2. Adsorption of Surfactant Materials

Surfactants are mainly synthetic chemicals. Surfactants are widely used in industrial cleaning and textile manufacturing, as well as in households. There are many types of surfactants, namely alkyl sulfates, linear alkyl-benzene sulfonates, alkyl ether sulfates, alkylphenol ethoxylates, alkyl ethoxylates, and quaternary ammonium halide compounds. While surfactants have significant applications, they can pose a threat to the aquatic environment when they exceed the desired concentration. Raw domestic wastewaters have linear alkyl-benzene sulfonate concentrations in the range of 0.54–21 mg/L [176], with toxicity to aquatic organisms when the concentrations exceed 0.1 mg/L. Surfactants can increase the chemical oxygen demand in municipal wastewater treatment, and hence can increase the solubility of other toxic organic compounds in soils [177,178].

The adsorption of the surfactants on the AC has occurred mainly through hydrophobic interactions between the AC surface and surfactants [179]. Few studies have been found with respect to the removal of different anionic and nonionic surfactants from wastewater using RH as adsorbent [180]. Specifically, RH can remove 243.9 mg/g of linear alkyl benzenesulfonate from aqueous solution, and the adsorption mechanism followed the Langmuir isotherm [181]. It has been found that lower pH adsorption of linear alkyl benzenesulfonate increases due to the weak basicity of linear alkyl benzenesulfonate [180]. Apart from the hydrophobic interactions, the van der Waals force between the surfactants and adsorbent also plays a vital role in the adsorption of surfactants on husk ash [180]. However, adsorption of different types of surfactants using RH and RH-derived AC could be further explored in more depth in order to check their suitability to broaden their applications. Finally, based on the maximum adsorption capacity value for the adsorption linear alkyl-benzene sulfonates, it can be predicted that similar kinds of surfactants, as well as anionic and nonionic surfactants, can be better removed using RH-derived AC, but further research is required.

### 4.1.3. Adsorption of Pesticides

With the advancement of modern agriculture, the use of pesticides is increasing day by day, resulting in increased water pollution. Pesticides are non-biodegradable, persistent, and carcinogenic in nature, and are thus considered a strong class of water pollutant. In addition, their toxicity together with their degraded products have adverse effects on the environment. Different adsorbents have been utilized for the removal of pesticides from water. For example, biochar has recently been utilized [182]. In addition, RH can be used for pesticide removal from water. Akhtar et al. [183] reported a higher removal percentage (over 98%) of pesticide triphosphate on RH. According to our best knowledge, RH-derived AC has only been applied in a few studies for the removal of pesticides. However, RH-derived AC could be a good solution for the removal of pesticides from water and wastewater in the future. As limited studies have been reported so far, in the future further applications can be extended using RH-derived AC for the removal of pesticides.

In summary, it was found that AC derived from RH has a great potential for the removal of various organic contaminants from water and wastewater using a very cheap and abundant material. The comparison data also show that ACs from RH have better performance than other CACs or other sources of ACs. Therefore, further research is needed on a commercial or pilot plant basis together with new potential applications (e.g., pharmaceutical and personal care products, and endocrine disruptor removal) of RH-derived AC.

## 4.2. Adsorption of Heavy Metals

Besides textile dyes, many heavy metals (HMs) are contained in water and wastewater, which have significant detrimental effects on humans as well as on aquatic species [184]. Different industrial activities and agricultural wastes are the main sources of HMs in the environment. HMs are the most hazardous contaminants in the aquatic and soil environments. The World Health Organization (WHO) has listed the most toxic metals as cadmium, chromium, copper, lead, mercury, and nickel [185,186]. AC is widely used for HM treatment [187,188]. Adsorption of HM ions from water is a straightforward method through electrostatic interactions [189]. Different factors such as the metal ion complex, surface area and porosity, the solution pH and the point of zero charges of the surface, the surface functionality, and the size of adsorbing species govern the adsorption of HMs in AC. Generally, chemically treated RH exhibited greater maximum adsorption capacities of HM ions than unmodified RH [190]. Suemitsu et al. [191] mentioned the better adsorption capacity of HMs. They used Procion Red and Procion Yellow-treated RH for the removal of Ni(II), Cr(VI), Zn(II), Cu(II), Cd(II), Pb(II), and Hg(II) ions from aqueous solution. They reported better removal efficiency for all HMs except Cr(VI), for which the removal efficiency was nearly 40%. In another study, activated RH exhibited appreciable adsorption (99 ± 0.5, 97 ± 0.6, 96 ± 0.8, and 95 ± 0.9% for the removal of Pb, Cd, Zn, and Cu ions, respectively) from low-concentration aqueous solutions [88]. Physically and chemically activated RH showed ~100% removal efficiency with initial concentrations of 9740, 540, 100, 30, 10, and 15 μ/L of Fe, Mn, Zn, Cu, Cd, and Pb, respectively, [61]. Hexavalent chromium has also been successfully removed by RH derived AC [192]. For removal of lead (II) ions, Raikar et al. [193] used RH in four different forms with or without chemical treatments. The maximum percentage removal of lead (II) ions was 93.36%, 94.8%, 96.72%, and 99.35% with RH, RHA, phosphoric acid-treated RH, and acetic acid-treated RH adsorbents, respectively. A comparison of HM removal by AC is presented in Table 8.

**Table 8.** Efficacy of CAC, RHA, and RH-derived AC for the removal of heavy metals (HMs).

| Adsorbate | Adsorbent Material | $C_0$ (mg/L) | Adsorption Capacity (mg/g) | Removal (%) | Ref. |
|---|---|---|---|---|---|
| Fe(III) | AC | 9.740 | | 100 | [61] |
| Fe(II) | CAC | 55.2 | 38.57 | | [194] |
| Mn(II) | AC | 0.54 | | 100 | [61] |
| Zn(II) | AC | 0.10 | | 100 | [61] |
| Zn(II) | RHA | 39.17 | | 96 ± 0.8 | [88] |
| Zn(II) | AC | 50 | 19.38 | | [195] |
| Zn(II) | RHA | - | 26.10 | | [196] |
| Zn(II) | RH | - | 29.69 | | [197] |
| Zn(II) | AC | 100 | | 75.1 | [191] |
| Zn(II) | CAC | 98 | 20.50 | | [194] |
| Cu(II) | AC | 0.03 | | 100 | [61] |
| Cu(II) | RHA | 40.82 | | 95 ± 0.9 | [88] |
| Cu(II) | AC | 100 | 29.00 | | [198] |
| Cu(II) | AC | 50 | 4.77 | | [199] |
| Cu(II) | AC | - | 112 | | [200] |
| Cu(II) | AC | 100 | | 78.8 | [191] |
| Cd(II) | AC | 0.01 | | 100 | [61] |
| Cd(II) | AC | 1000 | 20.24 | | [201] |
| Cd(II) | AC | 1000 | 16.18 | | [201] |
| Cd(II) | AC | 1000 | 11.12 | | [202] |
| Cd(II) | RHA | 39.87 | | 97 ± 0.6 | [88] |
| Cd(II) | RHA | - | 25.27 | | [196] |
| Cd(II) | RH | - | 21.36 | | [203] |
| Cd(II) | AC | 130 | | 99.2 | [191] |
| Pb(II) | AC | 0.015 | | 100 | [61] |
| Pb(II) | RHA | 39.74 | | 99 ± 0.5 | [88] |
| Pb(II) | AC | 400 | 108.00 | | [198] |

**Table 8.** *Cont.*

| Adsorbate | Adsorbent Material | $C_0$ (mg/L) | Adsorption Capacity (mg/g) | Removal (%) | Ref. |
|---|---|---|---|---|---|
| Pb(II) | RHA | - | 207.50 | | [204] |
| Pb(II) | AC | 120 | | 99.8 | [191] |
| As(III) | AC | - | 1.22 | | [173] |
| As(V) | AC | 0.09–0.85 | | 53 | [205] |
| Cd(II) | AC | 50 | 41.15 | | [98] |
| Se(IV) | AC | 50 | 40.92 | | [98] |
| Cr(VI) | AC | 250 | 48.31 | | [206] |
| Cr(VI) | AC | 150 | | 39.7 | [191] |
| Hg(II) | RHA | - | 46.14 | | [207] |
| Hg(II) | RH | - | 66.66 | | [207] |
| Hg(II) | AC | 200 | 384.62 | | [195] |
| Hg(II) | AC | 130 | | 92.7 | [191] |
| Ni(II) | RHA | - | 25.33 | | [196] |
| Ni(II) | RH | - | 8.86 | | [206] |
| Ni(II) | AC | 100 | | 61.6 | [191] |

In summary, it was found that AC derived from RH has great potential for the removal of HMs from water. The comparison data also show that AC from RH has better performance than other commercial ACs or other sources of ACs.

### 4.3. Adsorption of Inorganic Anions

Inorganic anions are another group of pollutants in wastewater which are known to be toxic and carcinogenic. The presence of these anions in ground and surface waters has resulted in severe contamination, and they can cause adverse health effects. For example, phosphate in surface water and groundwater caused water quality problems. Fluoride poses a serious threat to public health and causes dental and skeletal fluorosis. WHO limits the fluoride concentration in groundwater to 1.5 mg/L [208]. Also, water pollution occurs due to excessive discharge of nitrate ($NO_3^-$) and bromide ($Br^-$) in the environment [209–211]. Maximum limits of 50 mg/L $NO_3^-$ for adult and 15 mg/L $NO_3^-$ for infant drinking water are permitted [212].

RH-derived ash or carbon have been utilized for the removal of fluoride by Tantijaroonroj et al. [213]. It was reported that RH-derived ash could remove fluoride ions ($F^-$) by up to 42.5% at pH 2 (Table 9). RH was chemically and physically modified, and then the fluoride ion removal capacity was increased up to 75%, which is even higher than that of commercial AC (53.4%) [214]. Iron-impregnated activated silica carbon was produced from RH, and applied for the removal of fluoride ions by Majumder et al. [215]. They reported higher removal of fluoride ions. On the other hand, it was reported that the maximum adsorption capacity for bromate ion removal was 50 mg/L using granular AC [216]. Silver impregnation AC showed 85–93% bromide removal efficacy [217]. Other CACs were also applied for the removal of bromide ions, with satisfactory results. However, the removal of bromide ion using RH-derived AC has not been well established yet.

Two-fold enhancement of nitrate ions was achieved by the utilization of urea-modified RH-derived AC. The maximum nitrate adsorption capacity was found to be 8.11 ± 0.031 mg/g [218]. A 94.3% removal capacity of nitrate ion was observed for the anionic RH at optimized conditions (90 min, pH = 7.0) [219]. In addition, different ACs produced from sugar beet bagasse and coconut coir were also used for the nitrate removal [220–222]. However, the adsorption removal percentage of nitrate was similar to that of RH-derived AC [221]. In contrast 78–89% phosphate removal efficiency was achieved by the utilization of agro-waste RH at pH 6.0 with 2 h contact time [223,224].

**Table 9.** Adsorption capacities of inorganic anions by pristine RH, RH ash (RHA), and RH-derived activated carbon (AC).

| Adsorbent | Adsorbate | $C_0$ (mg/L) | Adsorption Capacity mg/g | Removal% | Ref. |
|---|---|---|---|---|---|
| AC | $F^-$ | 5 | - | 88.30 | [213] |
| RH | $F^-$ | 5 | - | 75 | [214] |
| AC | $BrO_3$ | - | 1.5 | - | [215] |
| AC | $Br^-$ | 10 | | 83.5 | [217] |
| AC | $NO_3^-$ | 15 | 3.76 | | [218] |
| RH | $NO_3^-$ | 100 | | 93.4 | [219] |
| GAC | $NO_3^-$ | 50 | 10.2 | - | [220] |
| AC | $PO_4^{3-}$ | 10 | - | 89.1 | [223] |
| AC | $PO_4^{3-}$ | 10 | - | 95.85 | [224] |

Therefore, AC has great potential in the removal of different inorganic ions from water and wastewater. RH-derived AC or carbons or ash have been well applied in the removal of fluoride and nitrate ions. On the other hand, data on the removal of phosphate ions ($PO_4^{3-}$) and bromide ions using rice husk derived ACs are limited. Therefore, further study should cover these gaps in order to check applicability on a larger scale.

*4.4. Gas Capture*

There are many toxic gases such as carbon dioxide ($CO_2$), carbon monoxide, ammonia, nitrogen dioxides, sulfur dioxide, hydrogen, and methane, among others. Among them, $CO_2$ is the most abundant gas in the atmosphere. It is a greenhouse gas [225–227]. There are many detrimental effects of those gases on the environment. Similarly, many methods have been adopted for the capture of those gases from the environment. For example, for the removal of $CO_2$, carbon capture and sequestration technology are mostly adopted. AC, especially RH-derived AC, has great potential for $CO_2$ adsorption due to its microporosity and the presence of nitrogen [228–230]. The $CO_2$ adsorption capacity of the hydrofluoric acid pre-dashed RH was 77.9 mg/g at 30 °C and 18.1 mg/g at 120 [231]. KOH-activated RH carbon was utilized for low-pressure $CO_2$ uptake, and maximum $CO_2$ uptake of 2.11 mmol/g at 0.1 bar and 0 °C was recorded. RH-derived AC exhibited a large $CO_2$ uptake of 6.24 mmol/g at 0 °Cat 1 bar [232]. Gargiulo et al. [233] overviewed $CO_2$ adsorption onto RH-derived sorbents and other materials under dynamic conditions. They mentioned that $CO_2$ adsorption onto those sorbents is driven by different factors such as pore volume, polarity, pore size distribution, surface area, and the presence of active sites i.e., functionality and unsaturated coordinative sites. On the other hand, Gansesan et al. [234] reported that RH-derived carbon can store gases (e.g., $CO_2$, $H_2$, methane) and also has electrochemical charge storage capacity. They mentioned the high adsorption capacities of those gases. For example, they reported values of up to 9.4 mmol/g (298 K, 20 bar), 1.8 wt % (77 K, 10 bar), and 5 mmol/g (298 K, 40 bar), respectively, for $CO_2$, $H_2$, and $CH_4$. These values were superior compared to many other carbon-based physical adsorbents. Finally, Dahlan et al. [235] used siliceous RH materials to produce RH ash, which was then applied for flue gas desulfurization in small-scale industrial boilers.

Therefore, RH-derived AC can be well applied for the removal of $CO_2$ as well as in desulfurization together with other uses. Limited data have been found for the removal of other gases (such as CO, $NH_3$, methane, and so on) using RH-derived AC. Hence, the applications of RH-derived AC can be broadened towards the removal of these gases.

*4.5. Air Cleaning*

There are many semi or volatile organic compounds (VOCs) which are very harmful for the environment. VOCs are mainly carbon-based chemicals which have high vapor pressure at ambient temperature. Many of them are very toxic, being lethal to humans and animals. Therefore, it is

highly necessary to reduce these effect in order to get clean air for a safer, smarter, and greener lifestyle [112,236–239]. Several filtration systems have been proposed but AC is widely used to adsorb those compounds from the air. AC derived from RH was found to have potential for the adsorption of VOCs. For example, KOH-activated RH-derived carbon showed higher sorption of toluene and phenol, with maximum adsorption capacity of 263.6 mg/g and 6.53 mg/g, respectively [240]. It was also found that post acid treatment of AC increased VOC adsorption capacity significantly [241]. In a different study, a relatively high phenol adsorption capacity (1919 mg/g) in the vapor phase was found by the utilization of hierarchically porous carbon derived from rice husk [112]. On the other hand, the adsorption of 16 VOCs (namely, n-pentane, n-hexane, cyclohexane, benzene, dichloromethane, trichloromethane, tetrachloromethane, 1,1-dichloroethylene, trichloroethylene, methanol, ethanol, 2-propanol, acetone, acetonitrile, diethyl ether, and ethyl acetate) in the gas phase was studied by Li et al. [238] using RH-derived AC. It was found that the presence of higher in the air decreased the adsorption capacity of VOCs; however, at lower humidity the adsorption of those compounds was higher [238]. Therefore, RH-derived AC can be a good option for future real-scale applications for filtering air pollutants. However, their potential mechanism of action as well as the reusability of AC should be considered carefully. In addition, more VOCs should be studied under experimental conditions in order to investigate the possible further applications of RH-derived AC.

### 4.6. Critical Assessment of Adsorption Research

The removal of contaminants from wastewater is a matter of concern to different stockholders. Among several methods, the adsorption method is being used extensively, and is a very simple, and effective process. Based on this principle, a large number of works have been devoted to the adsorption process using different adsorbents in both commercial as well as lab-scale experiments. However, adsorption studies mainly involve: (1) selection of a new adsorbent; (2) characterization; (3) physicochemical parameter studies; (4) experiments in batch or on a column basis with the adsorbent; and (5) fittings the parameters into different models and equations. In principle, any adsorbent should have some adsorption capacity into different contaminants. RH-derived AC has been found very effective in the removal of different organic and inorganic contaminants. Even RH-derived AC has excellent sorption performance over CAC. However, RH-derived AC has not been utilized for the removal of many contaminants. Hence, more research is required for a proper assessment of the efficacy of RH-derived AC.

### 4.7. Adsorption Mechanism of AC

Adsorption of different organics, inorganics, pesticides, surfactants, heavy metals, and so on onto adsorbent surfaces can happen through different mechanisms, namely hydrogen bonds formation, van der Waals forces, electrostatic interactions, surface precipitation, ion exchange mechanisms, cation exchange or anion exchanges, pore filling, π–π interactions, hydrophilic or hydrophobic interactions, diffusion processes, and so on. The removal of contaminants using an adsorbent usually follows four different steps such as (1) adsorbate transport to the adsorbent surface; (2) film diffusion onto the adsorbent surface; (3) adsorbate diffusion inside the pore; and (4) interaction between adsorbent pores and surface with adsorbate molecules. These kinds of interactions are very tough to control, but their interactions can be strong, weak, or more specific; this can be predicted and analyzed through different processes [242,243]. The potentially different adsorbents and different mechanisms to bind contaminants onto AC are shown in Figure 5.

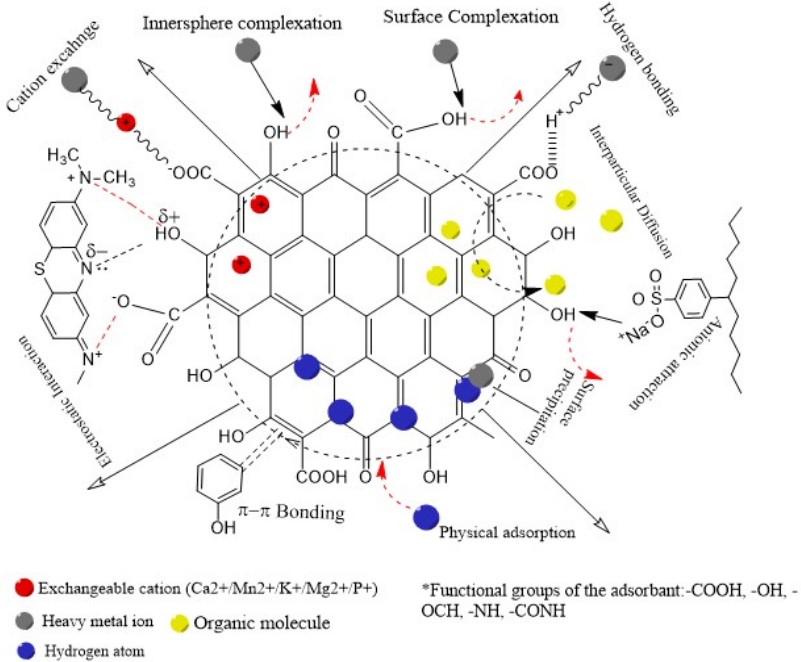

**Figure 5.** Adsorption mechanism of different types of pollutants on the AC surface.

From a mechanistic viewpoint, the rate-limiting step is considered as a critical factor in overall sorption processes [243]. Though different kinetics and equilibrium isotherm studies can help to identify the types of the adsorption process, in most cases, the mechanism of action needs to be predicted [244,245].

Cation and anion exchange occurs when opposite sites are present in both adsorbent and adsorbate molecules. In this case, the zeta potential value of the adsorbent is very important as it plays a vital role in the different ionic compounds. Hence, these kinds of bonds can easily form and quicken the overall adsorption process. For example, the removal of methylene blue by adsorption on the surface of AC occurs due to MB being in the $MB^+$ cationic form. The overall adsorption process follows the above mentioned four steps [246]. Similar kinds of different anionic and cationic interactions may occur between the adsorbate and adsorbent (Figure 6). The oxygen atom in AC can have strong effects on the adsorption of contaminants, especially when it presents to an edge of carbon surface [247,248]. Many studies showed that ion adsorption onto AC occurs due to the ion exchange with protons in oxygen functional groups [249,250]. According to metal ion classification, hard metal ions ($Zn^{2+}$, $Ni^{2+}$) are adsorbed to the surface of the functional group (-COOH, -OH). However, AC has π electrons on the surface of micropores, where the soft metal ions ($Pb^{2+}$) tend to be adsorbed [249,251].

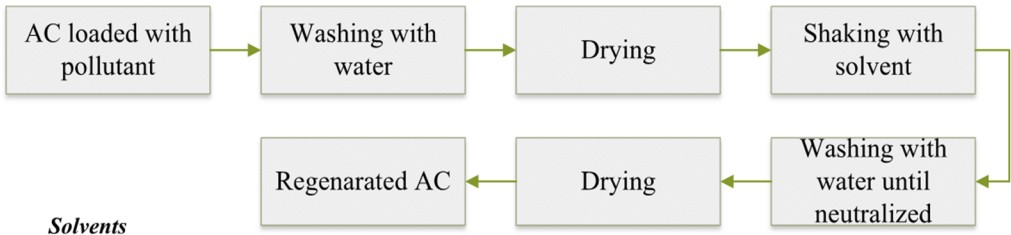

*Solvents*

- Organic pollutant (NaOH, HCl, methanol, ethanol)
- Heavy Metal ( HCl, $H_2SO_4$, $HNO_3$, KCl, $Na_2CO_3$ and EDTA)
- Phenolic compound, pesticides (Deionized water, $CaCl_2$ and NaOH)

**Figure 6.** A schematic on the desorption of solute and regeneration of AC.

It is common for water molecules to form different types of hydrogen bonds with surface oxygen groups or other functional groups of the adsorbents, where at least one hydrogen atom is present between the adsorbent and adsorbate molecules [252,253]. Hydrogen bond formation is very common for organic pollutants with the AC adsorbents.

Different types of π–π interactions (e.g., cationic or anionic and so on) can also occur within the adsorbent and adsorbate molecules in solution. π–π interactions belong to the noncovalent interactions which can contribute to chemical bonding, biomolecular structure formation, boiling points, solvation energies, and the structures of molecular crystals. For this kind of interaction, at least one of them should have π-electron rich or a deficient group in the form of a benzene or aromatic ring which can potentially cause these kinds of interactions. Surface functional groups in the adsorbate and adsorbent play a critical role in such kinds of interactions. Based on the type of surface functional group present in the structure and the solution pH, the adsorbent or adsorbate molecule either can act as an electron donor or electron acceptor site for electron–donor–acceptor interactions, electron acceptor–acceptor interactions, and electron–donor–donor interactions. Among them, electron donor–acceptor interactions are the strongest form of interaction. These kinds of interactions can significantly dominate sorption mechanisms [254].

Finally, some other weak bonds such as van der Waals forces (e.g., all weak interactions except the hydrogen bonds), surface deposition, and non-defined covalent bonds can also form between the adsorbent and adsorbate molecules.

### 4.8. Regeneration, Desorption, and Environmental Concerns of Used AC

#### 4.8.1. Regeneration

After the use of any sorbent, it is necessary to reuse it again and again for maximum profitability, check the extractability of solute, elucidate the mechanism of adsorption, and check reusability [255]. There are several methods for AC regeneration, namely solvent extraction [256] and oxidizing [257], thermal [258,259], microwave [260], and biological regeneration [261]. The solvent extraction process has been used for many years for the desorption of organic compounds using subsequent compounds such as chloroform, ethanol, dimethylformide, acetone benzene, and so on [256,262]. However, the advanced oxidation process has been focused on the regeneration of adsorbent [263]. In this vein, hydrogen peroxide ($H_2O_2$) and ozone ($O_3$) are mostly used to oxidize organic compounds by generation of hydroxyl radicals [264]. In comparison with other regeneration processes, the thermal process has some drawbacks. This is because the regeneration efficiency is sometimes hampered due to the formation of non-volatile compounds with higher molecular weight at the AC surface [257,265]. Pyrolysis temperature and duration also influence the weight loss percentage of AC [258]. Using microorganisms for the regeneration of AC with higher stability and prolonged duration of service is also recognized as a bioregeneration process [261]. Studies on the regeneration of spent adsorbent help to elucidate the nature of adsorption.

#### 4.8.2. Desorption of Organics and Heavy Metals

The desorption of the RH or RH-derived AC-loaded adsorbate (i.e., contaminants) allows the reuse and recovery of the adsorbent and adsorbate. For better results, parallel column use can facilitate adsorption and desorption processes. The desorption experiment depends on different factors such as the amount to be sorbed, the process requirements, and the economic considerations. The desorption process is usually carried out by mixing a suitable solvent with the contaminant-saturated adsorbent. The mixture is then shaken or centrifuge until the adsorbate desorbs from the adsorbent. For example, the maximum desorption values of textile dyes (Acid black 26, Acid green 25, and Acid blue) from pinecone were 93.16%, 26.97%, and 98%, respectively [266]. For dye to desorb from the adsorbent, hydrochloric acid is mostly used as it can easily desorb more than 90% of dye from adsorbent [267]. Thus, a stable original form of adsorbent is highly recoverable. A study on the adsorption/desorption of

benzene, toluene, ethylbenzene, and xylene (BTEX) suggested that the adsorption/desorption of BTEX was affected by chemical structure, solubility, and molecular weight. It is common that organic solvents such as methanol, ethanol, acetone, ether, and aldehyde be used for organic pollutant desorption from adsorbents. On the other hand, different studies suggest that acids like HCl, $H_2SO_4$, $HNO_3$, $Na_2CO_3$, and potassium chloride (KCl), bases like NaOH, and chelating agents like EDTA have an excellent capability for the desorption of metal ions [88]. Few have metals such as Cr (VI) and Ni (II) which can be desorbed using a basic medium. Figure 6 shows the desorption of solute and hence the solvent regeneration of adsorbent.

### 4.9. Environmental Concerns

Perhaps most of the criticism of AC is due to relevant environmental concerns. It is anticipated that contaminated-loaded AC needs to go through proper treatment processes before going to the environment. However, as we mentioned earlier that there are several ways for the desorption of contaminants (organics, inorganics, and so on) as well as several processes of regeneration. Even so, at the end-use of the sorbent, proper disposal of the adsorbent is necessary. As a part of this, recently thermal treatment processes (incineration, gasification, combustion, and so) have been applied for the conversion of adsorbent into alternative materials such as syngas through gasification (organic contaminant-loaded AC). On the other hand, HM-loaded AC can be first desorbed and then further applied in the soil in an area where metals needed for plant growth. Furthermore, the regenerated AC can be used for further applications such as fly ash, which can form a part of a cement composite or building materials like biochar (recently applied in building construction). Finally, as an alternative for the conversion of contaminant-loaded AC or for the use of desorbed and regenerated AC a new proper desorption treatment system that can desorb solute by up to 100% is necessary.

## 5. Catalytical and Energy Applications of RH-Derived AC

### 5.1. Catalytic Support

For carbon materials to act as catalyst support, the properties of high surface area and mesoporosity are necessary. Results show that mesopores in AC can significantly improve the catalytic activity of materials [266–270]. Lu et al. [271] produced porous carbon from RHs by $CO_2$ activation. Different transition metals can be immobilized or can be impregnated through different physical and chemical processes in order to produce different catalysts. Wang et al. [272] developed a method to produce high-quality biodiesel from soybean oil with an AC-based catalyst. The calcium oxide (CaO)/AC catalyst was used in the synthesis of fatty acid methyl ester. The best yield of biodiesel production was about 93.01%. Metal-impregnated and functionalized ACs were promising catalysts and adsorbents for various industrial applications as well. For instance, iron-containing catalysts had high catalytic activity and good effects for removing the harmful metal ion in wastewater. It was also reported that iron-containing AC formed an active and selective catalyst for phenol oxidation with $H_2O_2$ as oxidant [273]. It is revealed that iron displays better activity than other transition metals [274], zeolites, and other porous materials [274,275].

As AC from RH has a high surface area and a mesoporous and microporous structure, it can be an alternative catalytical material for various applications such as hydrogen production, oxygen evaluation reactions, $CO_2$ and CO reduction, and ammonia production.

### 5.2. Electrodes for Battery and Supercapacitors

AC is an amorphous material with a low presence of $sp^2$ carbon structures. Renewable biomass-derived AC has recently gained attention as a supercapacitor. From RH, hierarchical porous carbons with high surface area are highly possible. These hierarchical structures have excellent electrical double-layer storage capacity. Therefore, these kinds of AC have great potential as a supercapacitor. RH contains $SiO_2$ as a part of its components. $SiO_2$ is an insulator. Henceforth, after removing $SiO_2$,

the prepared AC from RH is mainly responsible for its conduction properties [276]. AC electron transport properties were investigated by Kennedy [276]. Liu et al. [277] mentioned that when the silica in RH is not removed, then KOH can react with the silica, which can hinder the formation of mesopores. Therefore, the obtained AC shows high microspore volume as well as a specific area up to 3263 $m^2$/g with a specific capacitance of 315 F/g at 0.5 A/g. Moreover, the carbon which was produced from RHs had a specific area of 2523.4 $m^2$/g. The specific capacitance of that carbon was 250 F/g at the current density of 1 A/g, which remained at 198 F/g when the current density was raised 20 times, indicating excellent rate performance. The carbon electrode produced from RHs also showed a long cycle life. The capacitance remained almost stable after the first ~100 cycles which kept stable up to 10,000 cycles [278]. This information clearly indicates the suitability of RH-derived AC to be used as a supercapacitor electrode.

Compared to CAC, RH-based carbons could provide higher double-layer capacitance [136]. Hierarchical porous carbon derived from RH was also used to enhance the electrochemical kinetics of the lead–carbon electrode [279]. Results showed that charge acceptance of the lead–carbon electrode is increased mainly due to the extra electrochemically active surface provided by AC.

RH-derived AC was also used in cathodes of Li–S batteries [280,281]. Amorphous nanoporous AC produced from RH showed a specific capacity of 730 mAh/g which remained at 140 mA h/g at 10 C (~3.75 A/g). After three cycles, the columbic efficiency was above 99% [282]. The high capacity, rate capacity, and long-term cycle life of the AC prepared from RH suggest promising applications to lithium-ion battery anode electrodes. Mai et al. [283] developed a highly porous AC with micro/mesoporosity through carbonizing RH with $K_2CO_3$. Elemental sulfur was uploaded to the micropores to obtain RH-derived AC (RHAC)@S composite materials. The discharge capacity was 1080 mA h/g at a 0.1 C rate after 50 cycles of charge/discharge tests at the current density of 0.2 °C. These results have demonstrated that the RH-derived AC is very promising cathode material for the development of high-performance Li–S batteries.

Therefore, RH-derived AC has great potential in the application in the field of batteries and energy storage capacitors and supercapacitors. Further study is required for the real-scale applications of RH-derived AC in the field of supercapacitors and batteries.

*5.3. Hydrogen Storage*

Very recently, hydrogen energy, as a renewable energy source, has come to the attention of many researchers. It has zero environmental emissions, as hydrogen fuel does not emit any toxic byproducts [284]. AC produced from RH has been utilized for hydrogen storage. For example, Chen et al. [38] evaluated the hydrogen storage capacity for two types of AC synthesized at room temperature from RH. Heo et al. [285] produced AC from RH using a KOH chemical activating agent, resulting in an increase in the surface area and pore size. The hydrogen storage capacity greatly dominates the nanometer size distribution and the microspore volume. Hence, hydrogen storage by RH-derived AC is in the preliminary stage, and it needs further study to obtain more data as well as to check the suitability of RH-derived AC for hydrogen storage.

*5.4. Energy Application*

Thermochemical methods like gasification, pyrolysis, combustion, cofiring, and biochemical processes are used for the conversion of biomass to biofuel [286]. The thermochemical process produces solid chars, liquid bio-oils, and syngas as the final products (Figure 7). Syngas usually contains carbon monoxide (CO), $CO_2$, $H_2$, and methane ($CH_4$), has relatively high higher heating values, and can be co-combusted with natural gases in a combustor [287]. The upgraded bio-oil can be used as the fuel for diesel engines [288]. The yield of pyrolysis product greatly depends on the feedstock type and operational conditions, and also on pyrolysis temperature [289].

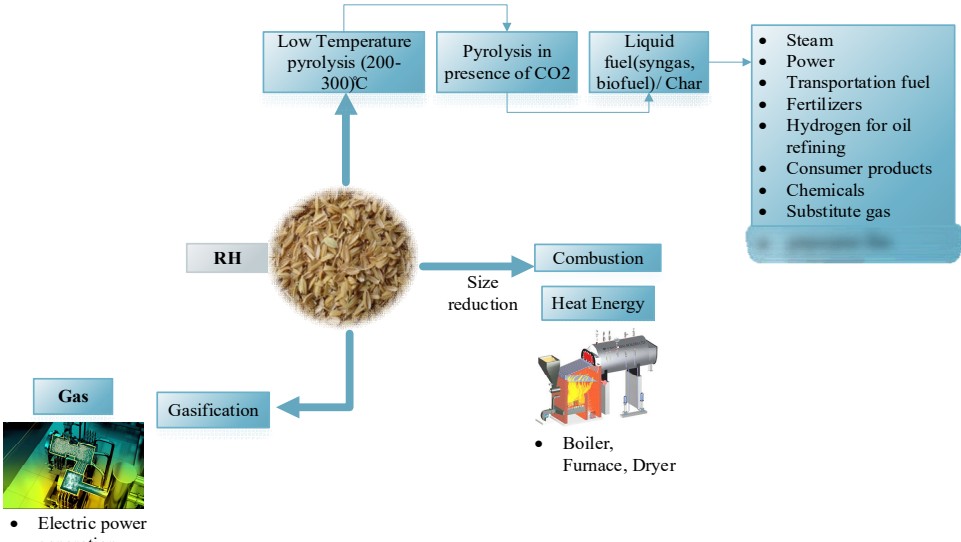

**Figure 7.** Possible energy applications of RH-derived AC and RH.

Islam et al. [290] reported a techno-economic analysis of the pyrolysis process for converting RH waste to pyrolysis oil and solid char. Williams et al. [291] pyrolyzed RHs in a fluidized bed reactor at temperatures of 400–600 °C. They analyzed the pyrolysis oils to determine their yield and composition in relation to process conditions. RH was paralyzed between 420 °C and 520 °C in a fluidized bed, and the chemical composition, heating value, stability, miscibility, and corrosion characteristics of bio-oil were determined [292]. Fast pyrolysis of RHs was carried out by Tsai et al. [293] in a fixed bed reactor using a pyrolysis temperature of 400–800 °C and a heating rate of 100–500 °C /min. Lu et al. [294] produced bio-oil in a fluidized-bed pyrolysis reactor and analyzed the bio-oil for its chemical and physical properties. Therefore, RH-derived AC has a wide range of energy applications which can further be considered in the future.

## 6. Future Prospects

For the adsorption of different organic pollutants, heavy metals, and inorganic anions from aqueous solutions, different methods have been adopted to prepare AC from RH. However, CACs are rather expensive, which is a big concern for real applications in water and wastewater treatment, particularly in developing countries. There are several main concerns that need to be addressed to utilize low-cost AC for the adsorptive removal of pollutants from water.

Future research should be focused on the following:

- This literature review indicates that more detailed systematic studies on the method of removing compounds as well as on technical improvements in the manufacture and use of adsorbents are required.
- Regeneration of used AC and proper disposal systems should be improved for zero environmental concerns.
- Most of the results are also based on laboratory benchmarks, suggesting that pilot-scale studies are needed to test the RH AC under actual field conditions.
- The applications of RH-derived AC need to be broadened with respect to different inorganic ions, HMs, organic contaminants (e.g., endocrine disruptors, pharmaceuticals, and personal care product removal), and gas capture (e.g., Mercury-Hg, sulfur dioxide-$SO_2$, CO, air pollutants).
- The mechanism of pollutant removal using HR-derived AC should be given more focus to obtain a clear understanding.
- The commercial production system should be developed in order to maximize the versatile advantages of its high adsorption capacity of different pollutants.

- A proper new regeneration new system needs to be developed.
- To address relevant environmental concerns, it is necessary to desorb the contaminants before dumping or further treatment of AC. Alternatively, suitable methods of disposal should be considered.

## 7. Conclusions

Adsorbents like AC with great adsorption capacity are promising materials for the future. Interestingly, the advantages of AC produced from RHs are comparable and, in some cases, greater than those of CACs produced from coal or other sources. Removal of silica from RHs is critical to the formation of porous structures in the activation process. However, the presence of silica can also affect the electrochemical performance of the resulting AC to be used as a supercapacitor. The presence of silica in RH could show higher performance for the adsorption of various pollutants. Therefore, based on the type of application, silica is either removed or remains unaltered. The chemical and thermal treatment of RH precursors to produce activated carbon leads to a higher adsorption of different pollutants. There are many shortcomings in the application of RH-derived activated carbons for the removal of many pollutants. Moreover, the mechanism of removal is not yet clear. Hence, further research is needed.

**Author Contributions:** M.M.A. and M.B.A. have collected data, designed the manuscript and written the main text. M.A.H.; M.D.H.; M.A.H.J.; J.H.; M.S.R.; J.L.Z.; and A.T.M.K.H. have designed, collected data and revised the manuscript. A.K.K. has supervised M.M.A. and monitored, revised, designed the manuscript. All authors have read and agreed to the published version of the manuscript.

**Funding:** This research received no external funding.

**Conflicts of Interest:** The authors declare no conflict of interest.

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
