# Peer review of "The Potentiality of Rice Husk-Derived Activated Carbon: From Synthesis to Application"

_processes, doi:10.3390/pr8020203_

Round 1
Reviewer 1 Report
The research exposed in this manuscript is included in a current and interesting topic: biomass by-product valorization for sustainable processes. Particularly, authors compile valuable and recent tendencies about activated carbon prepared from rice husk in terms of preparation methods and applicability. I recommend the publication in "processes" journal only after revising as listed below:
1- When precursor properties were described in section 2, authors list different factors that can affect the properties of the rice husk. Did authors observe any tendency related to the rice specie cultivated in those regions?
2- Section 4 can be improved by organizing pollutants by their nature. I mean, the sections for organic nature (contaminants, surfactants and pesticides) must be in one after another and same for inorganic nature compounds.
3- In line to previous point, table 6 can be improved by combination with table 7. Also, information about other contaminants of emerging concerns as pharma, personal care products, pesticides or surfactants would be valuable if possible.
4- Authors should include in table 8 information about inorganic anions removal.
5- Despite one of the main application of AC is pollutant removal from aqueous media, water is also required to regenerate the AC material. What is the real benefit of using these materials for water treatment if contaminated water is generated again in their reuse?
Author Response
We thank all of the reviewers and the editor for their constructive comments and suggestions to improve the manuscript quality. We have tried to address all of the concerns that were raised by reviewers. We have provided separate response (bold mark) for each comment or question or suggestion. We have also used track change mode in the revised manuscript to show the necessary changes. We hope revise manuscript will satisfy the reviewers concerns as well as will meet the journal required standard. Reviewer 1 Overall Comment: The research exposed in this manuscript is included in a current and interesting topic: biomass by-product valorization for sustainable processes. Particularly, authors compile valuable and recent tendencies about activated carbon prepared from rice husk in terms of preparation methods and applicability. I recommend the publication in "processes" journal only after revising as listed below: Response: Thank you very much for your constructive comments. We have revised the whole manuscript according to the suggestion of the reviewer. Question 1- When precursor properties were described in section 2, authors list different factors that can affect the properties of the rice husk. Did authors observe any tendency related to the rice specie cultivated in those regions? Response: Based on available literature, different parameters such as geological locations, biochemical analysis, proximate and ultimate analysis data have been listed in table 2 and explained in section 2. According to the reviewer suggestion, we tried to include specific data such as rice specie, but we didn’t find any relevant data/discussion on those references. So, we could not update the table according to suggestion of this reviewer, and we feel to keep the current context likewise as before. Question 2- Section 4 can be improved by organizing pollutants by their nature. I mean, the sections for organic nature (contaminants, surfactants and pesticides) must be in one after another and same for inorganic nature compounds. Response: Thank you very much for identifying the relevant issue. We value our reviewer concern, and we now have carried out the necessary changes in the revised manuscript i.e., we have moved surfactants and pesticides discussion subsections in the organic pollutants removal section. Therefore, subsections numbers together with references have been updated in the whole manuscript. Question 3- In line to previous point, table 6 can be improved by combination with table 7. Also, information about other contaminants of emerging concerns as pharma, personal care products, pesticides or surfactants would be valuable if possible. Response: We have tried to include more pollutants data on the removal of pharmaceutical and personal care products using only RH derived ACs. However, we did not notice such kinds of applications using RH derived ACs only. We believe that there are plenty of data for other types of ACs for the removal of pharmaceutical and personal care products together with endocrine disruptors. However, we put some concluding statements at the end of organic pollutants removal as well as in the future perspective sections. On the other hand, table 6 and table 7 contained different information’s, henceforth, we feel to keep the tables as like as before. Question 4- Authors should include in table 8 information about inorganic anions removal. Response: Thanks for the suggestion. According to the reviewer suggestion, we have made a new table (table 9 in the revised manuscript) for anions removal capacity and other parameters. Question 5- Despite one of the main application of AC is pollutant removal from aqueous media, water is also required to regenerate the AC material. What is the real benefit of using these materials for water treatment if contaminated water is generated again in their reuse? Response: It is very common that ACs materials are mostly used in aqueous media, and water media (with acidic or basic conditions) is also used for the regeneration of ACs materials. According to the general concept, regenerate solution is highly concentrated then the original concentration. So, based on different application, sometime, concentrated solution can be used for the recovery of the adsorbate. The main purpose of the AC regeneration is not to damp ACs again in the environment (more discussion on subsection 4.8). So, it determines the ability of adsorbent to use for many times. For the regeneration, higher or lower solution pH is normally used where adsorbate concentration is too high. So recovery chance is maximum. Now a day, there are lots of rare earth materials (e.g., Rubidium) have been extracted from the adsorbent. In that sense, it is normal to use another media to regenerate the adsorbent.

Reviewer 2 Report
The authors summarize recent progress on synthesis and applications of rice husk (RH) derived activated carbons, which could be interesting to the broad readership of the journal. Before considering accepting the paper, I would recommend the authors to address the following concerns:
The advantages of RH over other biomass in synthesizing activated carbons should be highlighted in detail. As an important approach in transferring biomass to carbon materials, hydrothermal treatment should be introduced in the paper. Air cleaning is one of the most important applications of activated carbons, which should be discussed in the paper. The discussion of “Gas Capture” in 4.6 was insufficient. The authors should give more details and examples. I would recommend the authors to add the following references, which are relevant to this paper.Chemical Society Reviews 39 (1), 103-116
Nanomaterials 2019, 9(1), 103
ACS Sustainable Chem. Eng. 2017, 5, 3087
Adv. Sustainable Syst. 2018, 2, 1700147
Author Response
We thank all of the reviewers and the editor for their constructive comments and suggestions to improve the manuscript quality. We have tried to address all of the concerns that were raised by reviewers. We have provided separate response (bold mark) for each comment or question or suggestion. We have also used track change mode in the revised manuscript to show the necessary changes. We hope revise manuscript will satisfy the reviewers concerns as well as will meet the journal required standard.
Reviewer 2
The authors summarize recent progress on synthesis and applications of rice husk (RH) derived activated carbons, which could be interesting to the broad readership of the journal. Before considering accepting the paper, I would recommend the authors to address the following concerns:
Response: Thank you very much for your constructive comments. We have revised the whole manuscript according to the suggestion of the reviewer.
Question 1: The advantages of RH over other biomass in synthesizing activated carbons should be highlighted in detail. As an important approach in transferring biomass to carbon materials, hydrothermal treatment should be introduced in the paper. Air cleaning is one of the most important applications of activated carbons, which should be discussed in the paper. The discussion of “Gas Capture” in 4.6 was insufficient. The authors should give more details and examples. I would recommend the authors to add the following references, which are relevant to this paper.
Chemical Society Reviews 39 (1), 103-116
Nanomaterials 2019, 9(1), 103
ACS Sustainable Chem. Eng. 2017, 5, 3087
Adv. Sustainable Syst. 2018, 2, 1700147
Response: Thank you very much. The advantages of RH over other biomass precursors have been added in the section 3.2 (track changes in the revised manuscript). In addition, we have now added hydrothermal treatment of biomass (added a new subsection 3.3.4 in the revised manuscript). Furthermore, according to the reviewer suggestions, we have updated the suggested references in the revised manuscript (subsection 3.3.4 in the revised manuscript). Finally, we have included more discussion related to other gases (e.g., SO2, H2 and methane) adsorption using RH derived ACs (subsection 4.6 in the revised manuscript). However, we did not notice any data on air cleaning gasses capture (i.e., benzene and xylene based air contaminants) by RH derived ACs although there are plenty of data on other types of ACs. Anyway, we have included them as future study gaps in the revised manuscript.

Reviewer 3 Report
Alam et al. reviewed on rice-husk derived activated carbon. They reviewed on a traits of rice-husk, methods for preparing activiated carbon from rice husk, applications and finally, future prospect on rice-husk derived activated carbon.
The overall oragnization of this review papaer is quite good and proper references are inserted finely.
Besides of aforementioned aspect, some minor revisions required.
Abbreviation should be refered at least once in main text before use it. The main text seems that delicate editing is needed.(such as different text size in line 91, wrong italic in line 390).
Author Response
We thank all of the reviewers and the editor for their constructive comments and suggestions to improve the manuscript quality. We have tried to address all of the concerns that were raised by reviewers. We have provided separate response (bold mark) for each comment or question or suggestion. We have also used track change mode in the revised manuscript to show the necessary changes. We hope revise manuscript will satisfy the reviewers concerns as well as will meet the journal required standard.
Reviewer 3
Alam et al. reviewed on rice-husk derived activated carbon. They reviewed on a traits of rice-husk, methods for preparing activated carbon from rice husk, applications and finally, future prospect on rice-husk derived activated carbon.
Response: Yes, we agree.
Question 1: The overall organization of this review paper is quite good and proper references are inserted finely.
Response: Thank you very much for your constructive comments. We have revised the whole manuscript according to the suggestion of the reviewer.
Question 2: Besides of aforementioned aspect, some minor revisions required.
Response: Thank you.
Question 3: Abbreviation should be referred at least once in main text before use it.
Response: We have now checked the whole manuscript to refer any abbreviation which has not covered in the last version.
Question 4: The main text seems that delicate editing is needed (such as different text size in line 91, wrong italic in line 390).
Response: Thank you very much to notice this issue. We have now revised the whole manuscript according to the suggestions of this reviewer.

Round 2
Reviewer 2 Report
The authors did not fully address the reviewer's comments.
(1) Several studies have reported the use of RH based carbon materials for air cleaning, for example, VOC adsorption and removal. The authors should include these studies in the paper.
https://www.sciencedirect.com/science/article/pii/S0301479719304694
https://www.sciencedirect.com/science/article/pii/S0960852419303724
https://www.sciencedirect.com/science/article/pii/S1383586616307274
(2) Some of the suggested references are not added in the revised manuscript.
Author Response
We thank the reviewer and the editor for their constructive comments and suggestions to improve the manuscript quality. We have provided separate response (bold mark) for each comment or question or suggestion. We have also used track change mode in the revised manuscript to show the necessary changes. We hope revise manuscript will satisfy the reviewers concerns as well as will meet the journal required standard.
Reviewer 2
Question-(1): Several studies have reported the use of RH based carbon materials for air cleaning, for example, VOC adsorption and removal. The authors should include these studies in the paper.
https://www.sciencedirect.com/science/article/pii/S0301479719304694
https://www.sciencedirect.com/science/article/pii/S0960852419303724
https://www.sciencedirect.com/science/article/pii/S1383586616307274
Response: Thank you very much. A new subsection (4.6 in the revised manuscript) has been created titled with “Air cleaning” in the revised manuscript. We have also added the suggested papers as well as more references in the revised manuscript.
Question-(2): Some of the suggested references are not added in the revised manuscript.
Response: We are sorry for the mistake. We have now added all of the references that missed in the previous version.
